**Supplementary Material for the paper titled**
**"On the (In)Significance of Feature Selection in High-Dimensional Datasets"**

Figure 43: SGD classifier results with ALL-AML microarray dataset. Trained and tested on 80:20 split, the plot shows that a random subset is able to match accuracy and AUC with all features, respectively.

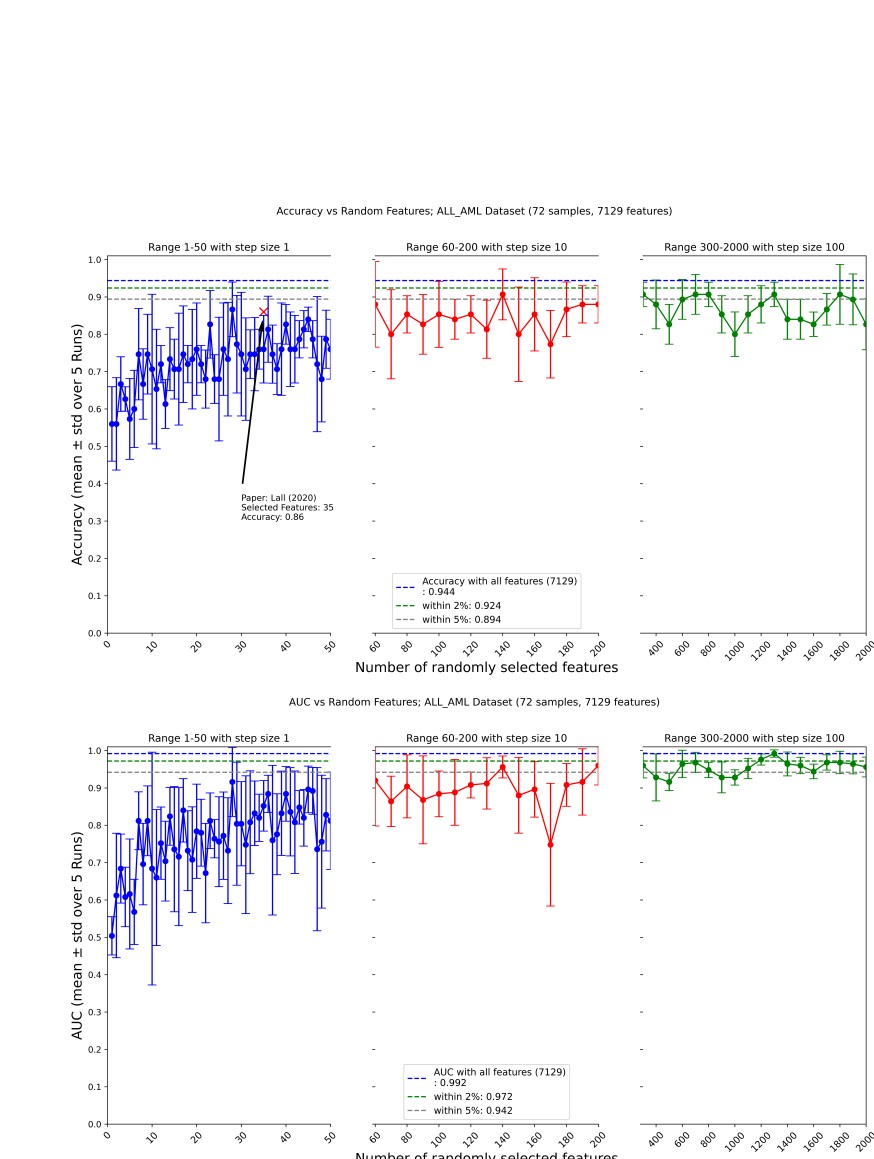

Figure 44: Support Vector Machine (SVM) results with ALL-AML microarray dataset. Trained and tested on 80:20 split, the plot shows that a random subset is able to match accuracy and AUC with all features, respectively.

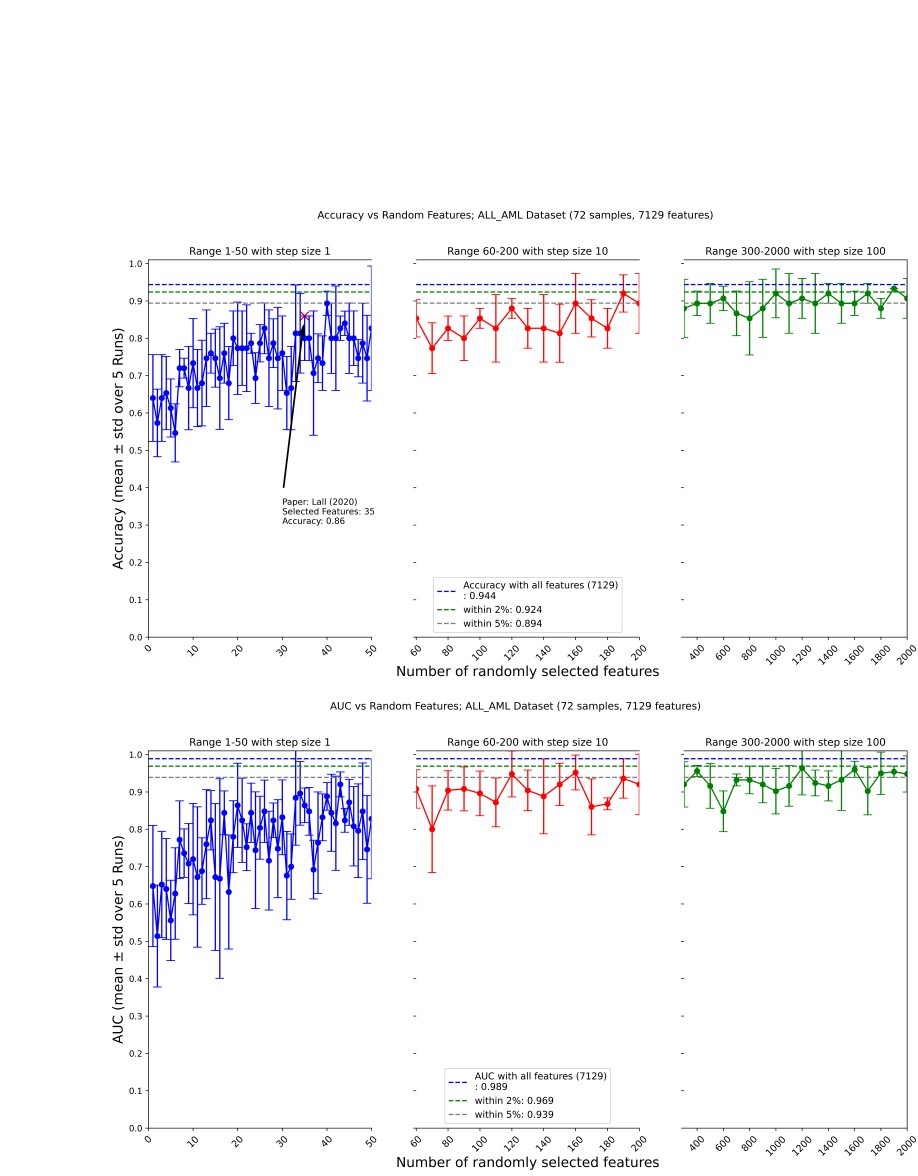

Figure 45: eXtreme Gradient Boosting (XGB) results with ALL-AML microarray dataset. Trained and tested on 80:20 split, the plot shows that a random subset is able to match accuracy and AUC with all features, respectively.

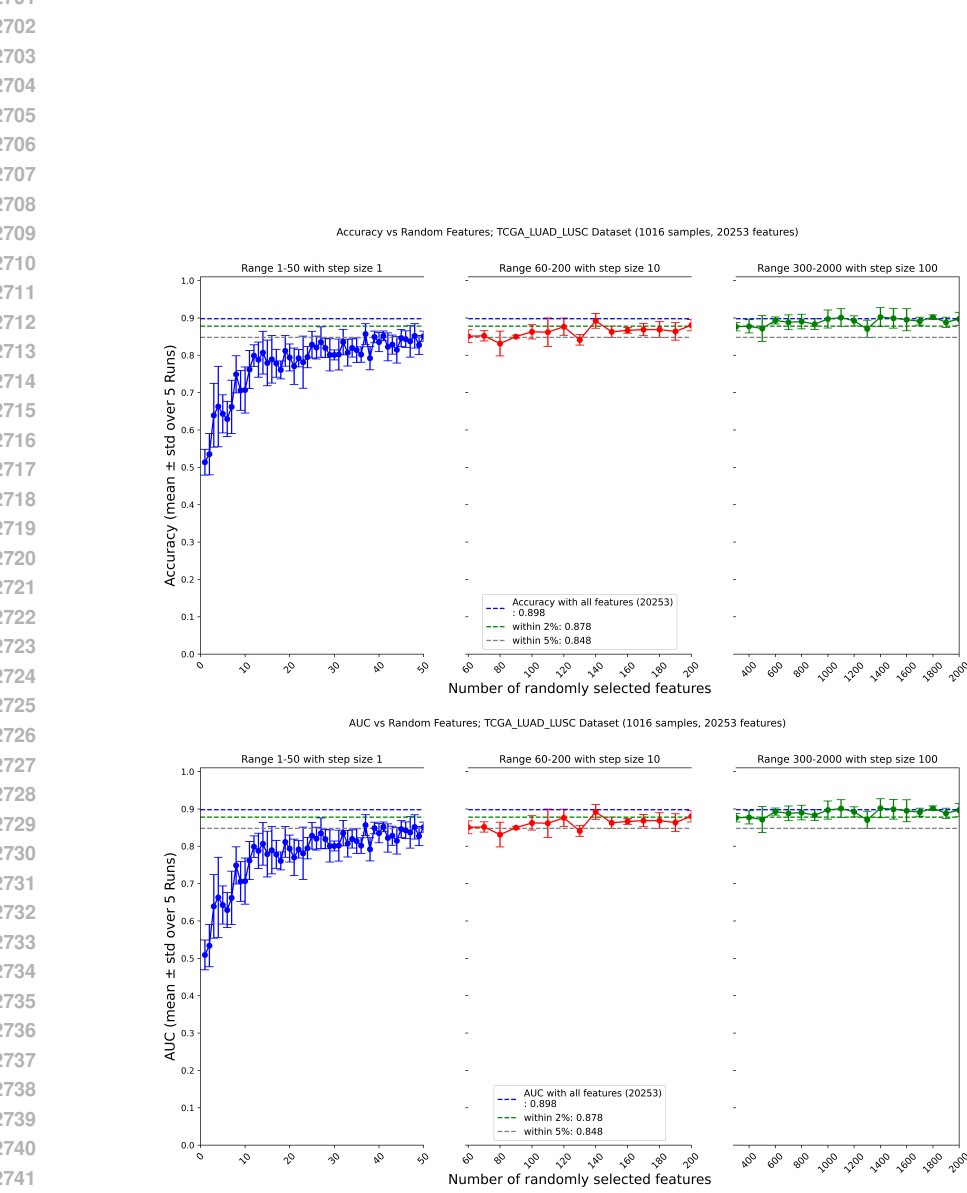

Figure 46: Decision Tree (DT) results with TCGA(LUAD/LUSC) bulk RNA-Seq dataset. Trained and tested on 80:20 split, the plot shows that a random subset is able to match accuracy and AUC with all features, respectively.

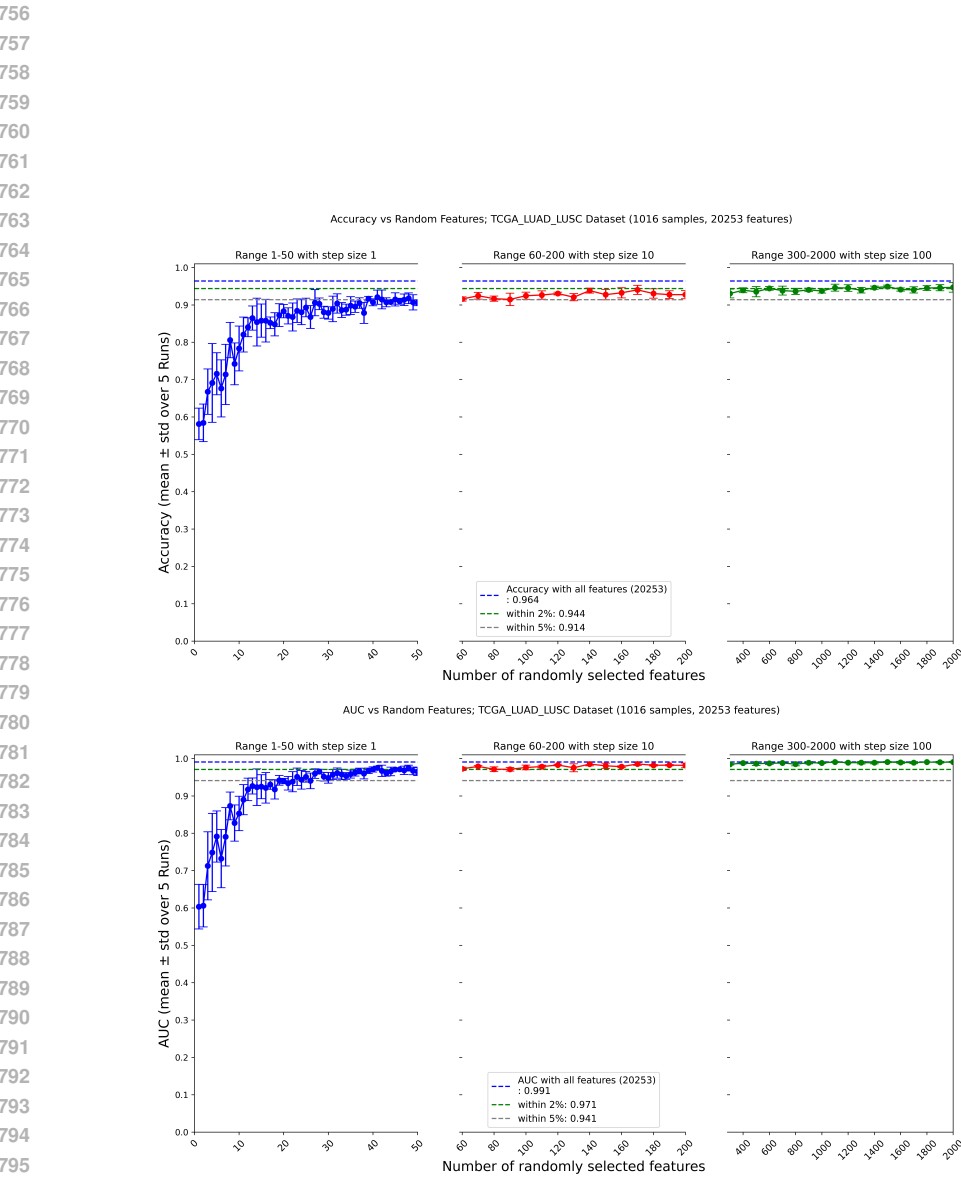

Figure 47: HistGradient Boosting (HistGB) classifier results with TCGA(LUAD/LUSC) bulk RNA-Seq dataset. Trained and tested on 80:20 split, the plot shows that a random subset is able to match accuracy and AUC with all features, respectively.

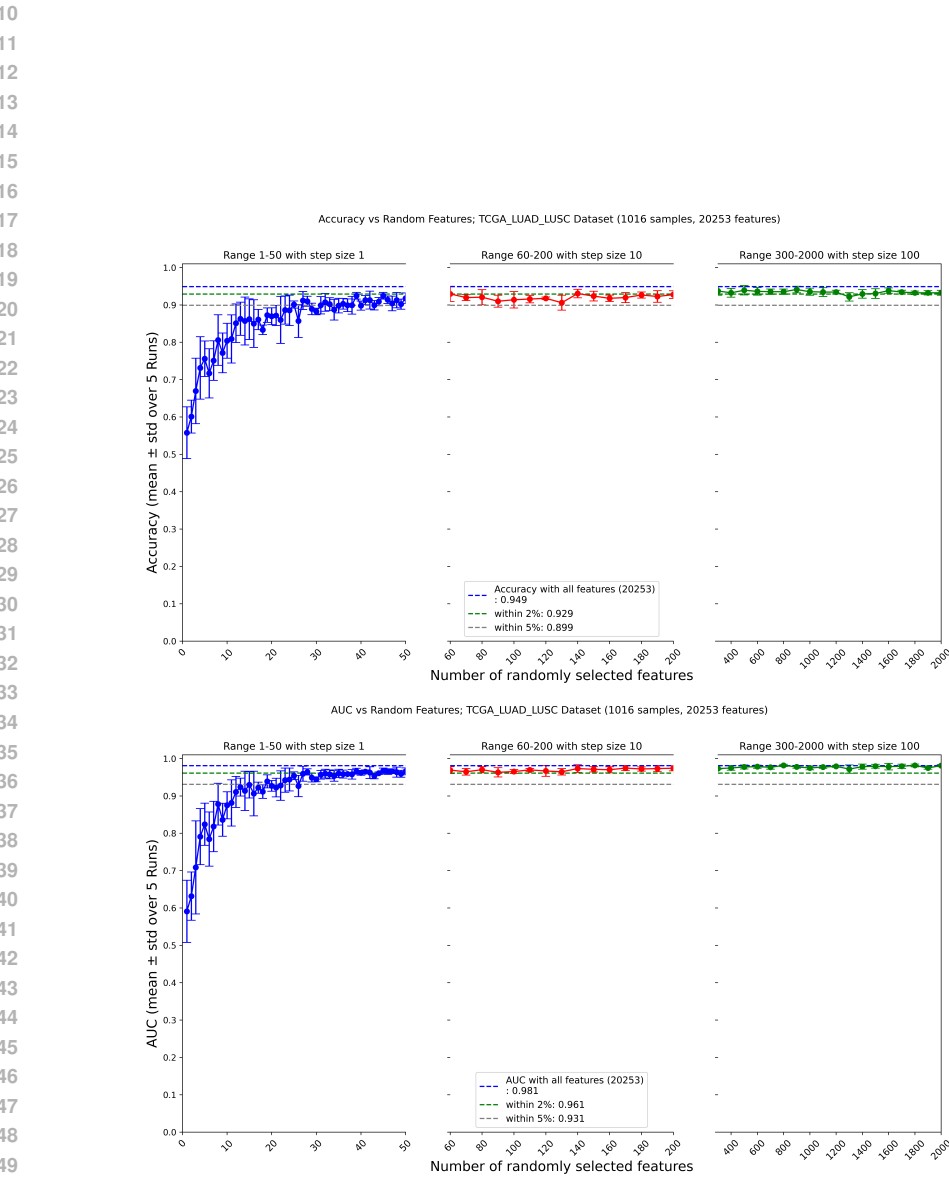

Figure 48: Logistic Regression (LR) results with TCGA(LUAD/LUSC) bulk RNA-Seq dataset. Trained and tested on 80:20 split, the plot shows that a random subset is able to match accuracy and AUC with all features, respectively.

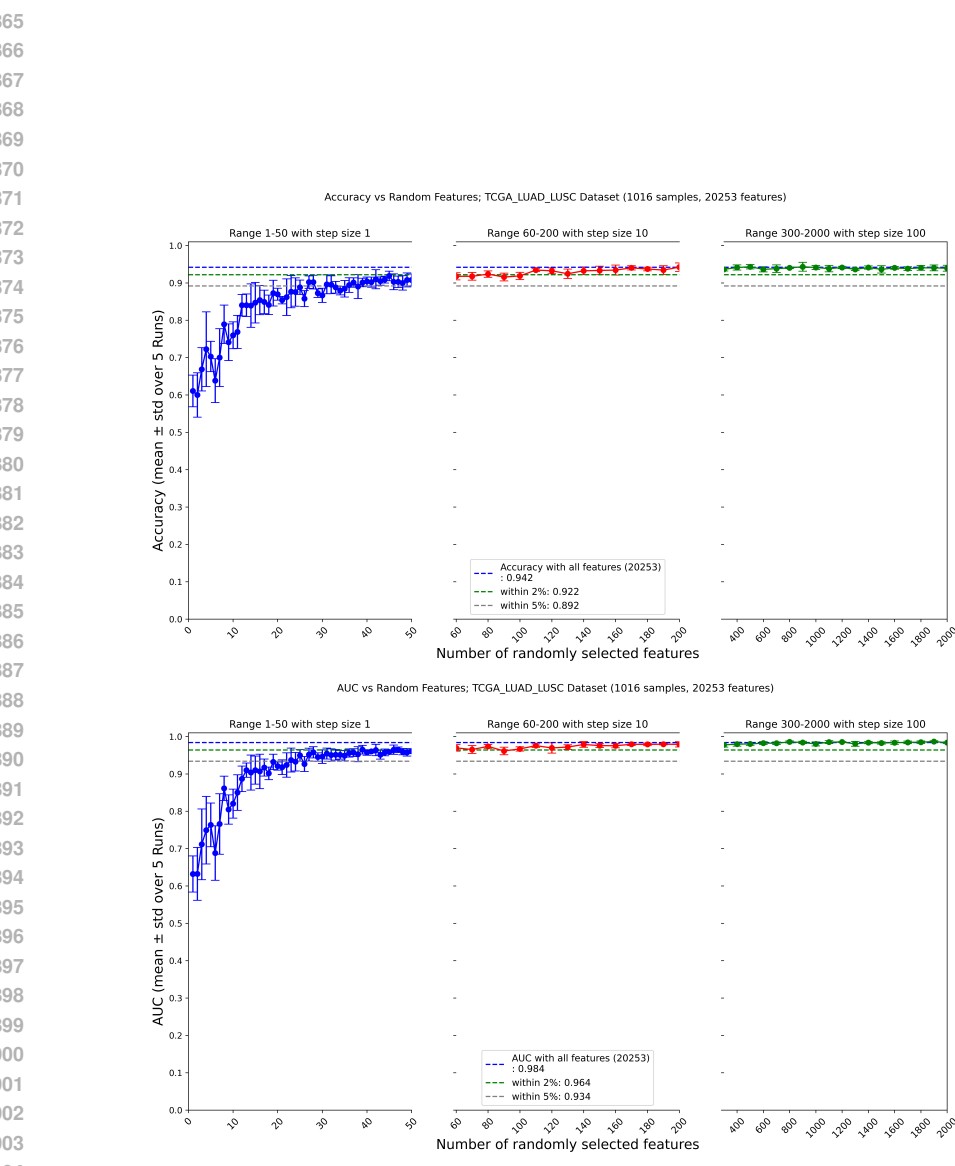

Figure 49: Multilayer Perceptron (MLP) results with TCGA(LUAD/LUSC) bulk RNA-Seq dataset. Trained and tested on 80:20 split, the plot shows that a random subset is able to match accuracy and AUC with all features, respectively.

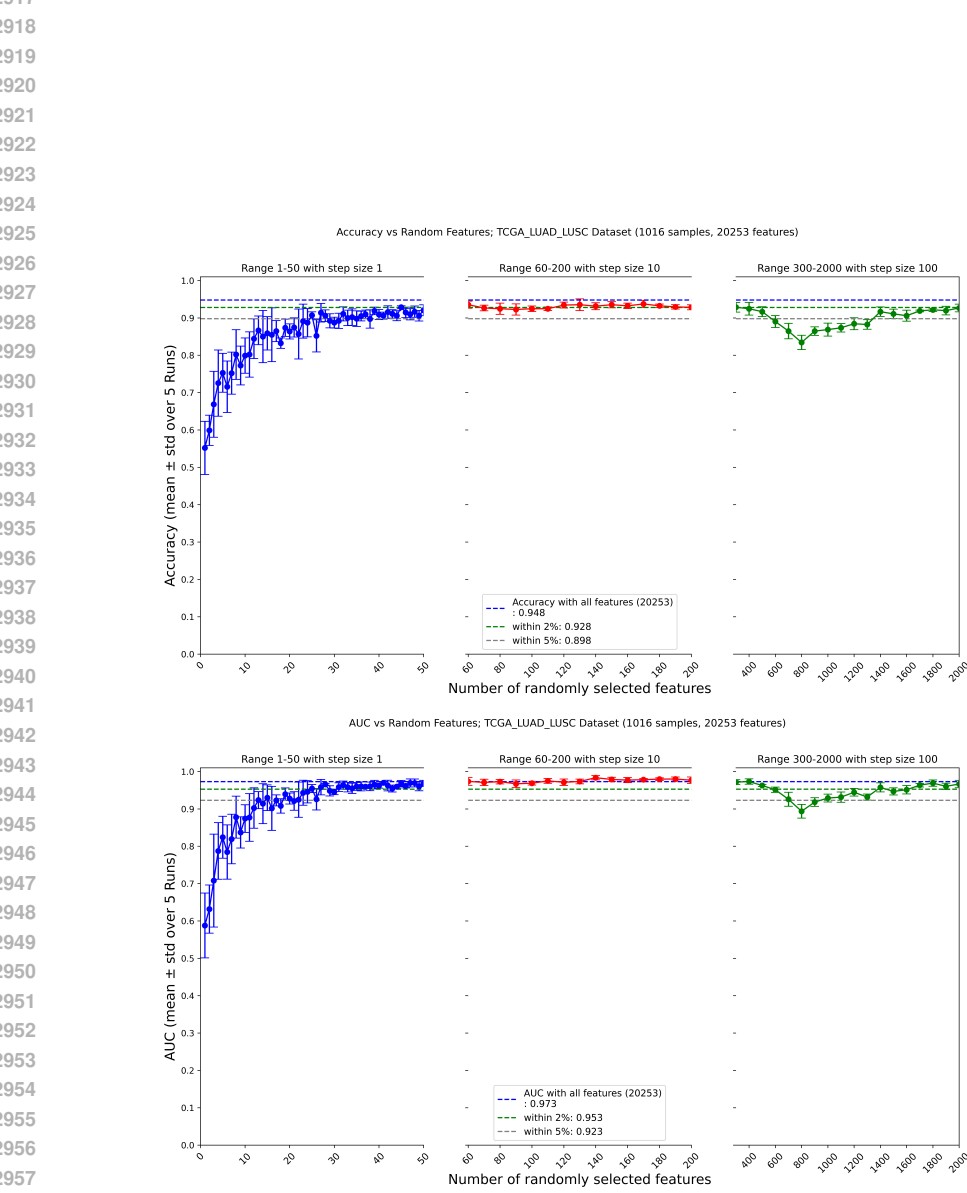

Figure 50: Ridge classifier results with TCGA(LUAD/LUSC) bulk RNA-Seq dataset. Trained and tested on 80:20 split, the plot shows that a random subset is able to match accuracy and AUC with all features, respectively.

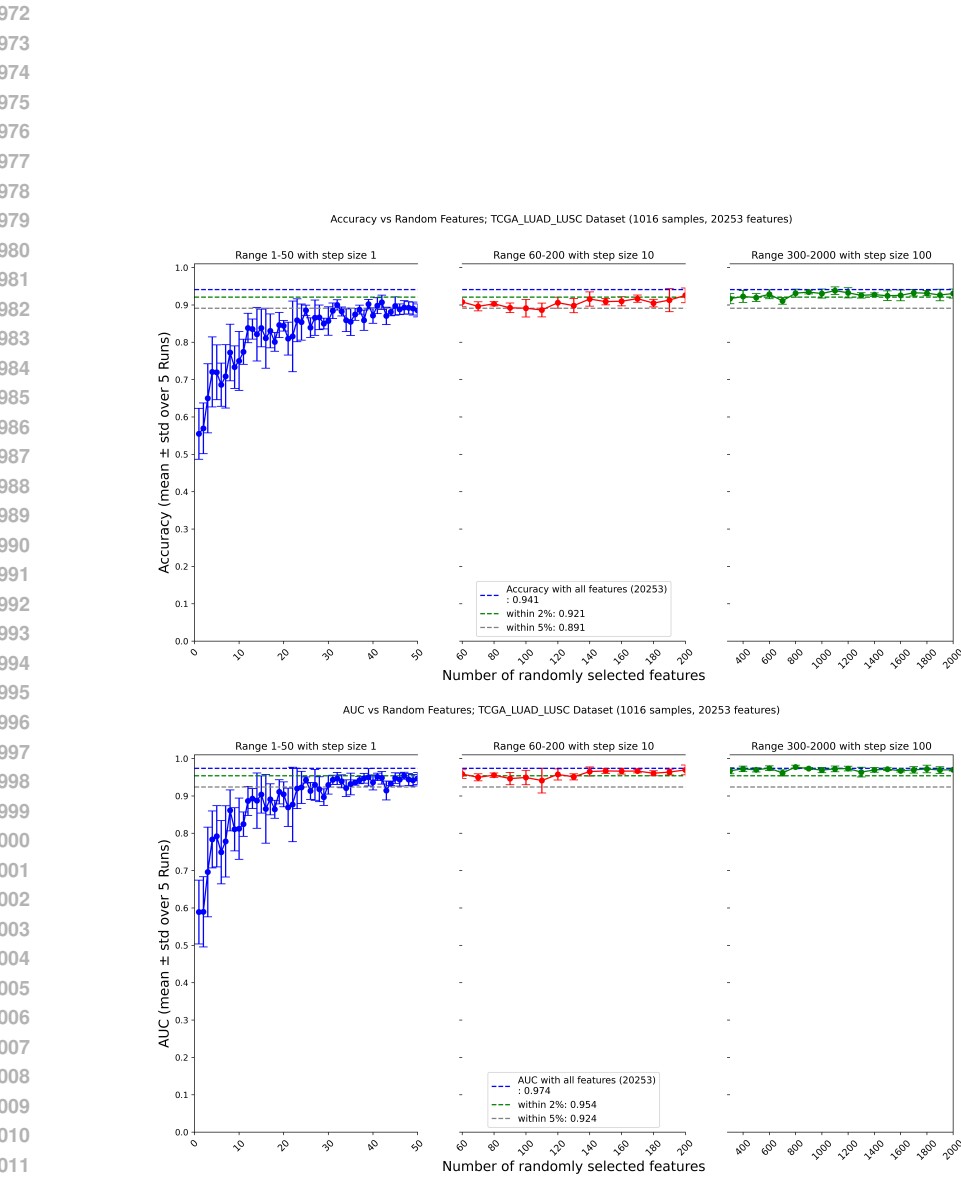

Figure 51: SGD classifier results with TCGA(LUAD/LUSC) bulk RNA-Seq dataset. Trained and tested on 80:20 split, the plot shows that a random subset is able to match accuracy and AUC with all features, respectively.

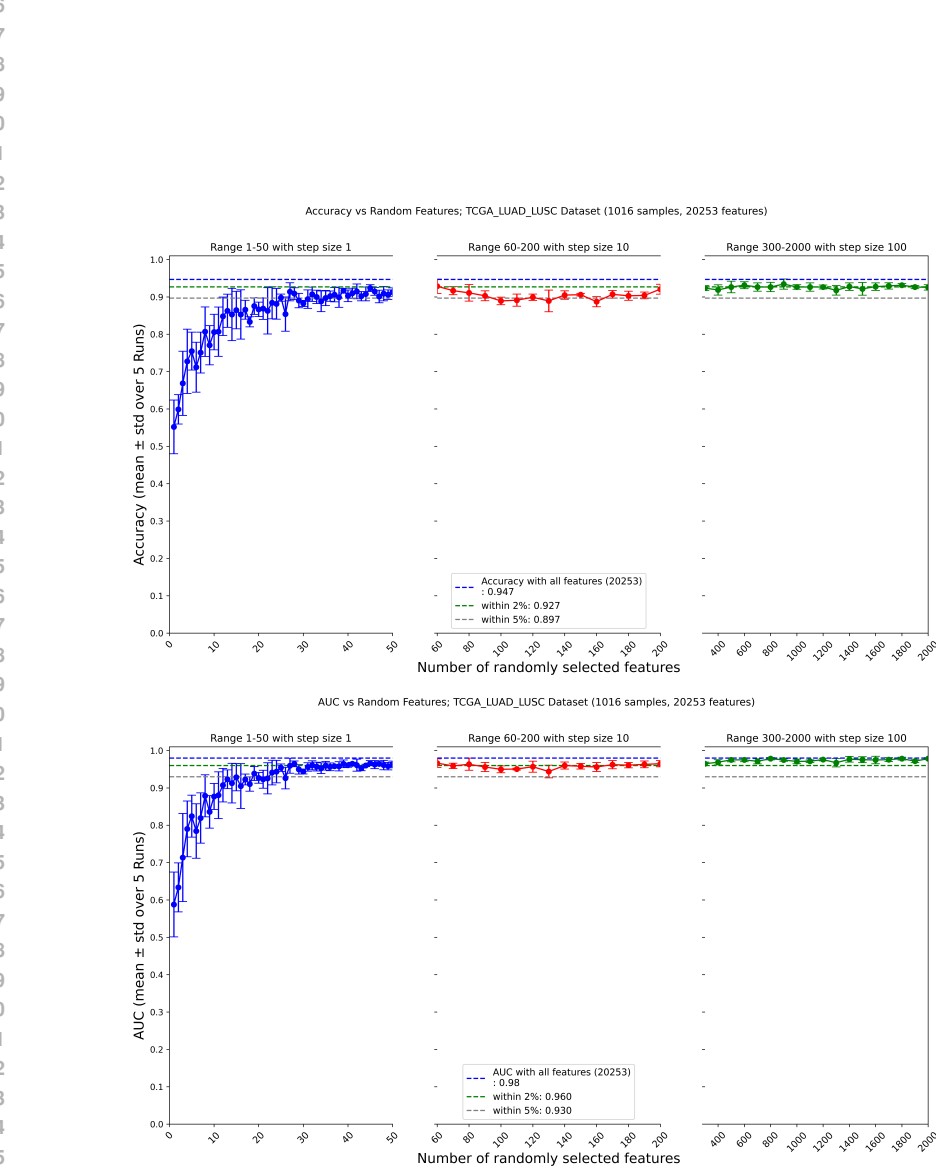

Figure 52: Support Vector Machine (SVM) results with TCGA(LUAD/LUSC) bulk RNA-Seq dataset. Trained and tested on 80:20 split, the plot shows that a random subset is able to match accuracy and AUC with all features, respectively.

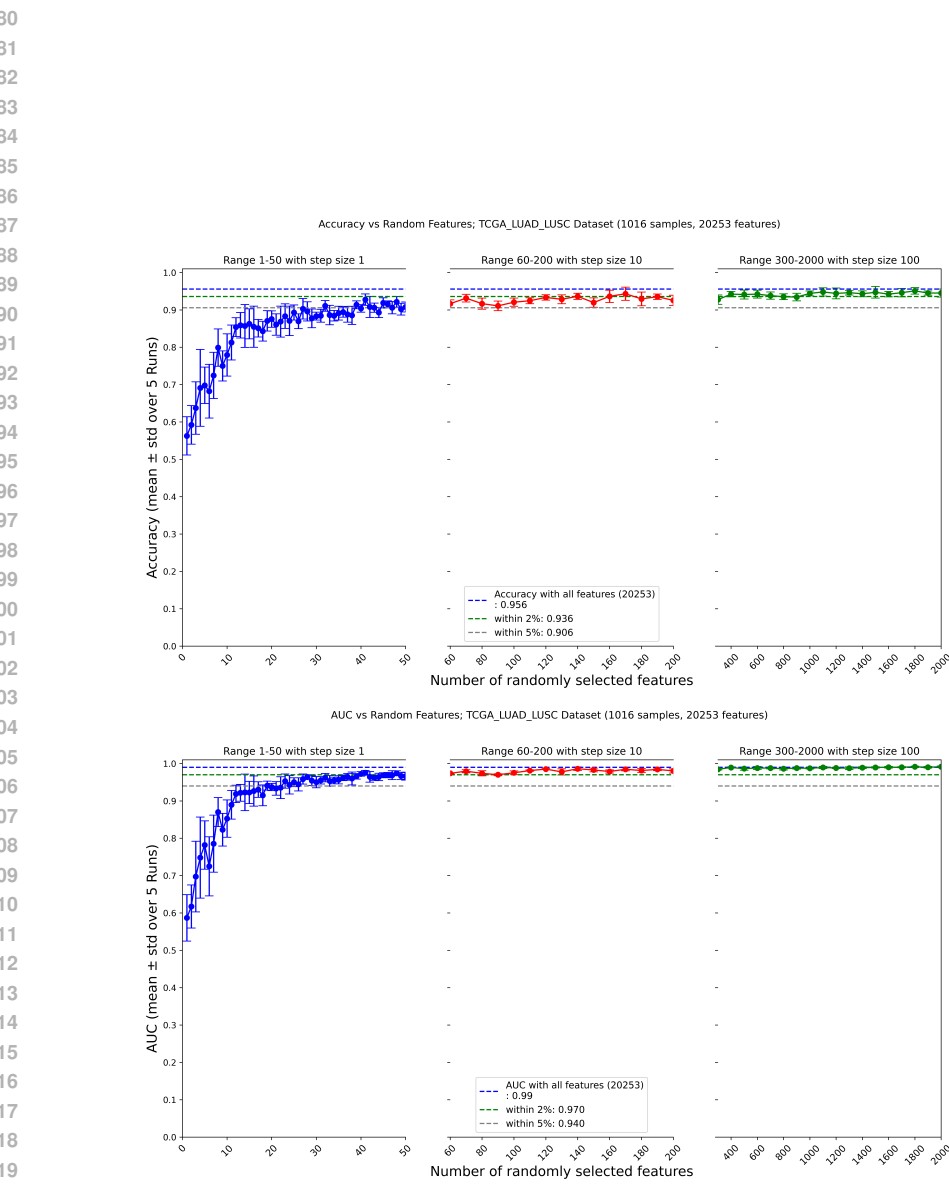

Figure 53: eXtreme Gradient Boosting (XGB) results with TCGA(LUAD/LUSC) bulk RNA-Seq dataset. Trained and tested on 80:20 split, the plot shows that a random subset is able to match accuracy and AUC with all features, respectively.