# OpenReview forum: "On the (In)Significance of Feature Selection in High-Dimensional Datasets"
_ICLR.cc/2026/Conference — Submitted to ICLR 2026_

### Official Review · Reviewer_AW9D · 2025-10-28

**Soundness:** 1
**Presentation:** 3
**Contribution:** 1
**Rating:** 2
**Confidence:** 3

**Summary:**

This paper studies feature selection for classification tasks. It uses established methods and metrics to perform feature selection on a long list of datasets, focusing on biological data (and microarray data within that).

The paper provides an empirical study on the effectiveness of randomly choosing a small, random subset of features from a given dataset for downstream applications (here, explicitly, classification). It claims that randomly selected features perform at least as well as 'cleverly' selected features. It also claims that a small, random subset of features can achieve the performance obtained by the full feature set and proposes that the minimum random set size that achieves this performance is a useful metric in describing the data.

**Strengths:**

- Reporting results on a long list of datasets. This is common in feature selection literature.
- Demonstrating with workflows implementing established methods
- Proposing a metric, minimum sufficient random sample size, that interpretably evaluates the collective strength of the features of a dataset

**Weaknesses:**

- The results do not seem to support a key claim of the paper (randomly selected features performing at least as well as cleverly selected features)
  - In Table 2, all values in column D are lower than the corresponding values in column A.
  - In Table 2, 2 out of 6 values in column E are higher than those in column A, but this is problematic:
    - The difference between columns D and E is that E uses an ensemble of classifiers. Column A could presumably also benefit from such ensembling.
    - The procedure for column E is not clear. The text says "ensemble of LR, RF, and XGB trained on different random subsets of features of same size as the published study" while the table says "same number of randomly selected features". If each ensemble uses a different random subset of the same size, then column E effectively uses up to 3 times more number of features. Alternatively if the total number of features available to column E is the same as that to column A, then it is not clear how the sizes of feature sub-subsets were allocated between LR, RF, XGB.

- With the exception of the Madelon and Gisette subsets, the sample count is less than the feature count for all studied datasets. (In Madelon, the paper's claim does not hold and all features are needed to achieve full performance. In Gisette, sample count and feature count are close to each other; 7000 vs 5000.) However, sample counts being much larger than feature counts is very common in modern datasets. For instance, it is nowadays routine to profile millions of cells with scRNA-Seq. It is possible that the trend the authors observe concerns primarily the #feature < #sample regime.

- The datasets are heavily skewed towards biological datasets. It is possible that the trend identified in this paper does not apply to all domains. I think the claim needs to be narrowed accordingly.

**Questions:**

- Could you try the proposed experiment on a larger scRNA-Seq dataset? Many such datasets are publicly available together with their cell type annotations, disease status, etc.

- (line 4141-416) This is confusing. Did the authors mean "The only three cases where **the full feature set** does better than chance..."? Table 1 does not show results for feature selection that is not random.

---

> ### Author Response · Authors · 2025-11-23
> **Response to Reviewer AW9D (part 1)**
>
> We thank reviewer for the detailed and thoughtful feedback. The comments helped clarify several points, and we address each item point-by-point below.
>
> **results do not support the claim—random features not as good as selected features (table 2, col A vs D)**
>
> Our claim is that any sufficiently sized random subset achieves performance comparable to selected features. The relevant comparison is the gap relative to the null baseline, not col A = col D exactly. if a feature selection ML system achieves 95% accuracy in some domain, it may seem to be good at first glance. However, if any random subset of features achieves, say (a) 93% accuracy, (b) 89% accuracy, (c) 50% accuracy, it would show the features and the results in a very different light. In the latter case, the FS seems to be achieving something useful and in the former cases not so much (though **b** suggests more significance for the selected features than **a**).
> In other words, in the 95% vs 93% case, the FS algorithm would not be "significantly" better than chance - we are using "significant” in the sense of passing/failing the null hypothesis. Our proposed metric, MSRSS, is an attempt by us to measure the natural underlying comparative number for a dataset to understand the significance of the selected features.
>
> We also highlight that any random subset performing comparably to the selected feature set from a published study is the main result of our paper. In addition to results in Table 2, for each dataset, we note that there is a size at which any random subset of that size can match the performance with selected features. The exact values per dataset are mentioned below:
>
> TCGA (LUAD/LUSC) - column D value is already within 1% of column A value, implying that any random subset of the same size as the published study (500 here) achieves similar performance (see figure 6).
>
> TCGA-pan cancer - any random subset of 200/20253 (\~1%) features achieves accuracy within 5% of the published feature set (see figure 26 in the Appendix B).
>
> ALL/AML -  any random subset of 300/7129 (~4%) features achieves ~91% accuracy (see figure 8).
>
> GSE4115 - any random subset of 100/22215 (0.5%) features achieves ~64% accuracy (see figure 20 in the Appendix B).
>
> These are the **“matching subset sizes”** we reference; **27 out of 30** datasets which we studied exhibit such points in their random-subset plots.
>
> **Column E > column A is problematic due to ensembling**
>
> In Table 2, main comparison is between column A and D. Column E just shows that with ensemble trained on completely random subsets we have more interesting results. All results presented in Table 1 and graphs are with one model only, not with ensemble.
>
> **Procedure for column E unclear (feature counts per model?)**
>
> We acknowledge that the way we presented our ensemble results may have been slightly confusing. We are reformatting and presenting our results (added as Table 6 in Appendix B) with some additional experiments to make things clearer.
>
> In the table below, column C has results for ensembles where each model is trained with a different random subset. However, each model in the ensemble is only trained on 1/n of selected feature set size (where n is the number of models in the ensemble, thus the total number of the features used by the ensemble is the same size as the set used by the published studies).
>
> Column D has results for ensembles where each model is trained with the same random subset - size equal to selected feature set size (as from a published study).
>
> Column E has results for ensembles where each model is trained with a different random subset - size equal to selected feature set size.
>
> | Dataset (selected feature count) | Results from Published Study | C: Different Random Subsets (1/3 size) | D: Same Random Subset | E: Different Random Subsets | Remarks |
> |--|--|--|--|--|--|
> | TCGA (50) | Li et al. 2017, 95.6% (50 features) | 85.76% | 87.72% | 92.77% | - |
> | TCGA (200) | - | 93.44% | 92.88% | 95.06% | one RF model with random 200/20253(~1%) achieves 91.0%, see Fig. 26 (Appendix) |
> | ALL/AML (35) | Lall & Bandopadhyay 2020, 86.0% (35 features) | 82.00% | 81.81% | 90.10% | one RF model with random 300/7129 (~4%) achieves ~91%, see Fig. 8 |
> | ALL/AML (51) | Golub et al. 1999, 98.6%; Cilia et al. 2019, 99.4% (51 features) | 84.48% | 90.29% | 90.10% | same as above. dataset has only 72 samples, making training challenging and thus a larger random subset is required to match |
> | ALL/AML (200) | - | 93.05% | 90.19% | 95.90% | same as above |
> | GSE4115 (5) | Zanella et al. 2022, 67.9% (5 features) | 57.72% | 52.97% | 64.71% | one RF model with random 100/22215 (0.5%) achieves ~64%, see Fig. 20 |
> | GSE4115 (50) | - | 67.31% | 69.46% | 69.49%|same as above. This dataset has only 187 samples making training challenging and thus a larger random subset is required to match|
>
> **(continued in the next comment)**

---

> ### Author Response · Authors · 2025-11-23
> **Response to Reviewer AW9D (part 2)**
>
> **feature count < sample count in modern datasets; effect may be limited to #feat > #sample regime**
>
> Given the extensive tests that we have done, we believe that our results hold for "tall” datasets with very large numbers of samples. In our paper, we already have the results for lung cancer sc-RNA-Seq dataset Bischoff2021 (21052 samples and 33514 features) - see Table 1 and Figure 3 E-H. We have also included the results for two breast cancer sc-RNA-Seq datasets, namely Wu2021 (96088 samples and 29733 features) and Qian2020 (15672 samples and 22276 features) from Weizmann 3CA repository. The random-subset phenomenon persists in these tall settings - see figure 31-24 in Appendix B.
>
> **datasets heavily biological; claim may not generalize**
>
> We agree and have adjusted the scope of the manuscript. the revised title is: “On The (In)Significance Of Feature Selection In High-Dimensional Biological Datasets” Our conclusions now explicitly apply to high-dimensional biological classification settings similar to the datasets examined, rather than the entire FS literature.
>
> **try experiment on larger scRNA-seq datasets**
>
> Results for the Bischoff2021 (Single-cell RNA-Seq lung cancer) dataset are already included in the paper as shown in Figure 3 E-H, where we show that any random subsets (20 such subsets) of 1200 features (∼3.6%) achieves AUC within 5% of the full feature set with remarkably low variance. This is surprising as this dataset contains 96.4% zeros (combined results for sparsity calculation are included in a table later.)
> Additionally, we have now added results for two breast cancer sc-RNA-Seq datasets, Wu 2021 and Qian 2020 from Weizmann 3CA. As it is a standard practice to use Highly Variable Genes (HVGs) to perform sc-RNA-Seq experiments, for these two additional datasets, we show results with HVGs. After selecting 5000 HVGs in Qian 2020, we perform our experiment with random subsets. Our results show that any random subset of ~100 features (2% of 5000) achieves within 5% accuracy (and AUC) of 5000 HVGs.  We find similar results for the Wu 2021 dataset with 3000 HVGs selected (due to paucity of time we repeat each experiment for 3 runs instead of 20 earlier). All the plots are now included as figure 31-34 in Appendix B.
>
> **line 4141–416: confusing phrasing**
>
> We thank the reviewer for catching this inadvertent error. We have now updated the manuscript accordingly.

---

> > ### Comment · Reviewer_AW9D · 2025-11-27
> >
> > Thanks for the responses. However, it solidified my understanding of the manuscript as detailed below.
> >
> > - The response conflates significance and effect size. It is very possible that 95% accuracy is statistically significantly better than 93%.
> >
> > - To claim statistical significance, p-values should be reported. A non-parametric shuffle test would be a good alternative: out of 1000 randomly chosen subsets, how many achieve or beat the performance reported in column A (not just "come close")?
> >
> > - “size at which any random subset of that size can match the performance with selected features”
> >     - The argument similarly suffers from lack of statistical rigor. Coming within 1% is not a meaningful result. Can the authors perform a proper statistical test and report a p-value?
> >     - The paper's argument centers on small subsets explaining the data well. This is different from any subset of the same size explaining the data as well as an optimal subset (in a statistical sense). The phenomenon that small subsets can explain the data well is already well-known and behind all the work on feature selection and sparse sensing. The common step of aggressive dimensionality reduction is also related to this phenomenon.
> >
> > - In a similar manner, the authors use the phrase “null hypothesis”. This use needs to be accompanied by formal statistical tests, p-values, and a clear definition of that null hypothesis.
> >
> > - I’m a bit confused regarding MSRSS. It is mentioned both in the Intro and under Contributions. However, there is no quantification regarding that. Conclusions section has one sentence on it (“Additionally, our MSRSS metric appears to have some interesting properties (e.g., it remains remarkably stable across totally different models - the reasons for this are unclear).”) MSRSS certainly conveys something about the effect size, but I don’t see how this captures significance.
> >
> > - Procedure for Column E: Thanks for the clarification. This seems to be in line with my previous understanding and the ensemble column in the original submission effectively has access to more features.
> >
> > - “Column E just shows that with ensemble trained on completely random subsets we have more interesting results.” I’m not sure what interesting means here. I don’t see how it supports the main claim. Again, my general understanding is that the paper suffers from a lack of statistical rigor.

---

> > > ### Author Response · Authors · 2025-12-03
> > > **On statistical significance, null hypotheses, and random-subset evaluation**
> > >
> > > We think there is a communication gap here: we are, in fact, doing pretty much exactly what the reviewer is suggesting. As the reviewer correctly identified, there needs to be a way to gauge significance which isn’t just looking at the numbers and the effect size, and that way is to compare the results against those from random feature subsets, with some “n” repetitions. Then, one should try to find something like a p-value.
> > >
> > > Unfortunately, the nature of machine learning does not readily allow us to transfer the concept of a p-value into the current scenario; further, FS is also trying to find the correct cardinality feature set (perhaps just adding 1-2 more features might increase the performance sufficiently to be considered a superior option). So, as the reviewer re-discovers, we need to run multiple random-feature-set tests at various sizes, and see how the accuracy increases, and how the FS results stack up against it. This is literally what we do - it’s pretty much the whole point of our paper. We then find that a lot of existing studies do not perform well when this measure is applied. Prior work does not do this - it compares against the whole feature set. The point is that one needs to compare the FS outcome with this random feature set curve.
> > >
> > > Classical or frequentist probabilities give meaningful error bars only when the data-generating process is well-specified and iid sampling holds. In our setting, this is not the case: machine learning in high-dimensional biological datasets do not admit mechanistic probabilistic guarantees about “feature importance”, “significance,” or future generalization. Thus the only falsifiable benchmark is randomized baselines. These baselines measure exactly how much predictive signal one can obtain without any assumptions or clever choices. If a feature-selection method cannot outperform random subsets under identical evaluation conditions, then its claimed “importance” is not empirically distinguishable from chance - we believe that this null hypothesis has been stated multiple times in the paper (and even in the abstract).
> > >
> > > Regarding the ensemble related results, we have added a more detailed table to avoid any ambiguousness about the number of features being used in each experiment.
> > >
> > > We agree that various studies have shown that multiple signatures exist - as mentioned in other replies, our point is universality, not just multiplicity: random feature subsets of surprisingly small size perform comparably to curated FS (and all features), challenging assumptions about utility of FS in high dimensional biological datasets. To summarise, in high-dimensional settings where no method can offer concrete probabilistic guarantees about future performance or feature “importance,” randomized baselines provide the only verifiable standard; our results show that many feature-selection methods fail to clear even that bar.

---

### Official Review · Reviewer_49XN · 2025-10-31

**Soundness:** 1
**Presentation:** 1
**Contribution:** 2
**Rating:** 2
**Confidence:** 5

**Summary:**

The paper presents a large-scale empirical analysis challenging the assumption that feature selection improves predictive performance in high-dimensional datasets. Across 30 datasets, mostly gene expression microarray data focused on cancer the authors show that small random subsets of features (0.02–1%) consistently match or outperform models trained on full feature sets or published feature selections.

**Strengths:**

The authors address an important question regarding the utility of feature selection in high-dimensional datasets, a topic with significant implications for machine learning and computational biology.

**Weaknesses:**

The paper has several limitations that reduce the strength of its conclusions:
1) The dataset selection process is not described. The authors provide no inclusion or exclusion criteria, making it unclear how the 30 datasets were chosen.
2) The dataset pool is heavily biased toward cancer-related gene expression studies from the Gene Expression Omnibus (GEO), yet the conclusions are generalized to the entire field of feature selection.
3) Cancer and inflammation datasets are known to display large-scale, disease-specific expression changes, which increases the likelihood that randomly selected genes still carry predictive signal.
4) Training on all available features may introduce substantial noise, artificially lowering performance and making small random subsets appear stronger than they are.
5) The paper overlooks much of the recent literature on feature selection and related theoretical analyses.

**Questions:**

I would ask the authors to address the weaknesses above:
1) Please specify the inclusion and exclusion criteria for dataset selection. For the microarray datasets in particular, were they drawn from a defined subset of GEO, or selected manually from thousands of available series?
2–3) The analysis should be extended to more diverse domains. Even within transcriptomics, including single-cell RNA-seq datasets would provide a stronger test case. In single-cell data, subtype classification often depends on a few key marker genes, making it unlikely that random subsets would perform comparably.
4) Please include actual feature selection algorithms in the experimental comparisons rather than relying solely on published feature sets.
5) I recommend consulting prior work on the multiplicity of genomic signatures, which offers theoretical grounding for the observed phenomena:
https://journals.plos.org/ploscompbiol/article?id=10.1371/journal.pcbi.1000790
https://jmlr.org/papers/v14/statnikov13a.html
https://link.springer.com/article/10.1007/s10618-020-00731-7

---

> ### Author Response · Authors · 2025-11-23
> **Response to Reviewer 49XN**
>
> We appreciate reviewer's careful reading and thoughtful comments. we respond to each concern point-by-point below.
>
> **dataset selection criteria**
>
> We thank the reviewer for this comment. We should have included our selection criteria. We have added a dedicated subsection 3.2 titled “Dataset Inclusion Criteria”.
> We include datasets only if they satisfy the following:
> 1. a reasonable number of samples (at least 100)
> 2. a large number of features (at least 2,000)
>
> plus datasets which are in widely cited feature-selection studies even if 1 and 2 above are not satisfied.
>
> **bias toward cancer / generalization**
>
> We agree and have adjusted the scope of the manuscript. the revised title is: “On The (In)Significance Of Feature Selection In High-Dimensional Biological Datasets” Our conclusions now explicitly apply to high-dimensional biological classification settings similar to the datasets examined, rather than the entire FS literature.
>
> **large-scale cancer expression changes and random subsets**
>
> We appreciate this concern. However, our results do not rely on selecting random subsets that happen to overlap with “true” disease-driven features. Across datasets, any sufficiently sized random subset performs comparably, even when subsets are selected with or without replacement. This suggests a universal, distributed signal rather than accidental correlation with a specific set of relevant genes. Because the expected performance is stable across random subsets, it is unclear which particular subset should be chosen as a reference for correlation analysis, and this is precisely why we emphasize **universality** rather than **multiplicity**.
>
> **full-feature baseline may be noisy**
>
> All previously published FS studies in this domain seem to compare their results against the full feature set: (e.g.) a stereotypical paper might say “The full feature set of 10000 features achieves 93% accuracy whereas we achieve 92.5% accuracy with only 100 features.” Alternatively, they compare against previous studies- which in turn at some point end up comparing against the full feature set. This, we believe, is an incorrect baseline or null hypothesis: if the claim is that a study has cleverly found 100 important features, the appropriate comparison should be against a “non-clever” selection. We compare random subsets with both the full feature set and the feature selected subset (Of course, the full feature set is the limiting case for the random subset.)
>
> **missing theoretical connections**
>
> We thank the reviewer for highlighting this literature. we now cite and discuss these works in section 2 - Related work.
> While signature multiplicity theory shows that several distinct but causally meaningful gene sets may achieve similar predictive performance, our results go further: the expected accuracy of any random feature subset is comparable to curated or algorithmically selected features across many datasets. Following on from the language of the cited papers, we do not just claim multiplicity of features, but universality: that it not just some features that are correlated but that any random subset of features (of sufficient size-which is surprisingly small, comparable to the FS cardinalities in published studies) contain enough information. This indicates that predictive signal is highly redundant and distributed - to the point where feature selection may provide little practical advantage in high-dimensional biological classification tasks. Thus our results are stronger than presented in these papers. To the best of our knowledge, previously no one has ever shown a universal random subset is good enough - there is no cleverness involved (which is the whole point, making the FS results fail in terms of significance vs the null hypothesis.).
>
> **including actual feature-selection algorithms**
>
> We thank the reviewer for the suggestion. We have added common FS techniques (lasso, elastic net, rf-importance) across datasets where corresponding published results exist. These new comparisons are in appendix A.
> |Dataset|Published Result| Lasso| Elastic Net | RF-Imp | Remarks |
> |----|----|-----|------|------|---|
> |TCGA (LUAD/LUSC, 500) | Chen & Dhabi 2021, 94.2% (500 features) | 95.48% | 95.28% | 95.18% | Random 500/20253 (2.5%) achieves 93.6%, see Fig. 6 |
> |TCGA (50) | Li et al. 2017, 95.6% (50 features) | Results Pending |Results Pending |Results Pending | Random 200/20253(~1%) achieves 91.0%, see Fig. 26 (Appendix) |
> | ALL/AML (35)|Lall & Bandopadhyay 2020, 86.0% (35 features) | 95.81% | 94.38% | 98.57% | Random 300/7129 (~4%) achieves ~91%, see Fig. 8 |
> | ALL/AML (51)|Golub et al. 1999, 98.6%; Cilia et al. 2019, 99.4% | 95.71% | 93.05% | 97.14% | Same as above. Despite advancements in feature selection methods, results from Golub 1999 still hold strong- suggesting the robustness of the method used.|
> |GSE4115 (5)|Zanella et al. 2022, 67.9% (5 features) | 62.11% | 66.27% | 67.31% | Random 100/22215 (0.5%) achieves ~64%, see Fig. 20|

---

> > ### Author Response · Authors · 2025-11-24
> > **Response to Reviewer 49XN - Continued**
> >
> > **including single-cell RNA-seq datasets**
> >
> > Results for the Bischoff2021 (Single-cell RNA-Seq lung cancer) dataset are already included in the paper as shown in Figure 3 E-H and Table 1, where we show that any random subsets (20 such subsets) of 1200 features (∼3.6%) achieves AUC within 5% of the full feature set with remarkably low variance. This is surprising as this dataset contains 96.4% zeros (combined results for sparsity calculation are included in table 4 in Appendix A.) Additionally, we have now added results for two breast cancer sc-RNA-Seq datasets, Wu 2021 and Qian 2020 from Weizmann 3CA. As it is a standard practice to use Highly Variable Genes (HVGs) to perform sc-RNA-Seq experiments, for these two additional datasets, we show results with HVGs. After selecting 5000 HVGs in Qian 2020, we perform our experiment with random subsets. Our results show that any random subset of ~100 features (2% of 5000) achieves within 5% accuracy (and AUC) of 5000 HVGs. We find similar results for the Wu 2021 dataset with 3000 HVGs selected (due to paucity of time we repeat each experiment for 3 runs instead of 20 earlier). All the plots are included as figure 31-34 in Appendix B.

---

### Official Review · Reviewer_1ZQK · 2025-11-01

**Soundness:** 3
**Presentation:** 3
**Contribution:** 3
**Rating:** 10
**Confidence:** 4

**Summary:**

This is a refreshingly honest paper that questions the validity of results published in 30 biological studies and this paper shows that randomly choosing a small collection of features and training a simple ensemble model with Linear Regression, Random Forest and XGBoost, gives comparable results to the published results!

**Strengths:**

ML as a community needs more such papers that challenge the conventional norm.  The results are quite powerful and a bit surprising. Having analyzed many of these datasets, I couldnt but wonder why others havent published such papers earlier.

A biologist can have one of two reactions to the claims in this paper: "wow, this is surprising/shocking" or "the analysis is flawed because of ..."! For either of these extreme reactions, this paper may be worth accepting to provoke deeper discussions. I actually think ICLR is the wrong venue for this paper for the visibility but the authors should have sent this to Nature/Science sub-journals.

**Weaknesses:**

The paper may have flaws in the analysis but I think the authors have been honest about their analysis and opened up their code and tried to validate them independently.

**Questions:**

A biologist can have one of two reactions to the claims in this paper: "wow, this is surprising/shocking" or "the analysis is flawed because of ..."! For either of these extreme reactions, this paper may be worth accepting to provoke deeper discussions. I actually think ICLR is the wrong venue for this paper for the visibility but the authors should have sent this to Nature/Science sub-journals.

If you want to cherry pick specific issues in the analysis, here are some trivial basic questions:
a. When they randomly select a sub-set of features, did the features they picked have strong correlations with the features deemed relevant by the published paper. If even a few features have strong correlations, then the random selection defeats the purpose.

b. Did the authors check how was the original datasets generated? Was the single cell RNA or bulk RNA data imputed using any zero imputation algorithm before the dataset was published? This completely changes the equation. Many imputation algorithms used in the literature give different imputations on the same data with limited agreement in the generated matrix. This could completely change the results.

c. Did the authors check if the generated data had any notion of experimental validation of gene types or cell types in the sequencing methodology?

d. Did the authors check the number of zeroes in the matrix? Many raw RNA matrices have more than 50-95% of the features to be zero before imputation. So if you randomly choose 0.5-2% of numbers, most of the numbers may be zero. If this is not the case, then there may be serious issues from the imputation pipeline.

e. Why did the fraction of randomly sampled features change from paper to paper? Are the authors optimizing their parameters to get the best outcome result for each dataset? That would constitute cheating in the analysis. They should clarify the same.

f. Many of the published results typically highlight very few edges as "outcome edges" that have significance from their analysis. Now, are the authors are comparing against the right metrics reported from these papers. The 95+% accuracy with a limited set of features published in these papers seems to imply that many of these papers are reporting several features. Most Nature/Science/Cell/Immunity papers focus on "one inference edge" in a paper. Something seems off in the way the results are reported.

Despite all these questions, I do love this paper and I think ICLR should accept such papers even if the results are flawed! Papers that challenge conventional wisdom are much harder to publish and should be encouraged.

---

> ### Author Response · Authors · 2025-11-23
> **Response to reviewer 1ZQK**
>
> We sincerely thank the reviewer for the generous and deeply engaged assessment. The encouragement, as well as the pointed questions, helped us refine the writing and strengthen our methodology. We especially appreciate the reviewer’s recognition of the broader impact of examining these assumptions, and we are grateful for the care and openness reflected in their comments. We have incorporated all clarifications into the revised manuscript (highlighted in blue color).
>
> We are answering all the questions here in the order they were posed:
>
> **a. on correlation between randomly selected features and published-relevant features**
>
> Ans: We show that any random subset (selected with or without replacement) can match performance with all features. In cases where we find published results, we show that with a certain size of random subset (0.02%-1%) FS performance can be matched within 5%. So we are arguing for universality and not just multiplicity of correlations. In addition, if any random subset can achieve comparable performance, a question arises, against which of these random subsets we do the correlation study? Because any such random subset works (results for 20 randomly selected subsets show this), correlation between these random subsets and the features deemed relevant by the published paper is not significant in this study.
>
> **b. imputation in the original datasets**
>
> Ans: None of the 30 datasets (our sources include CuMiDa, TCGA, Weizmann 3CA and NIPS 2003 FS challenge) were imputed before download or by us. Also, we checked that no zero imputation was done for ALL/AML dataset from Golub et al. 1999 and Colon dataset from Alon et al. 1999.
>
> **c. experimental validation of gene or cell types**
>
> Ans: Several studies on microarray data, including Golub et al. (1999) on ALL/AML, Li et al. (2017) on TCGA, Chen & Dhahbi (2021) on TCGA lung cancer, zanella et al. (2022) on GSE4115, Cilia et al. (2019) on ALL/AML, and Lall & Bandyopadhyay (2019) on ALL/AML, evaluate selected genes only through computational metrics. None of these works performs downstream biological or wet-lab validation of the identified features. As a result, their reported gene sets serve primarily as algorithmic artifacts rather than experimentally validated biomarkers. This gap motivates analyses validating the assumed biological relevance of selected genes, especially in high-dimensional settings where random subsets often perform comparably. Thus published gene sets are algorithmic outputs, not validated biomarkers, so our null-model comparison is appropriate.
>
> **d. zero counts and sparsity**
>
> Ans: We report dataset sparsity using three metrics: global fraction of zero entries, median fraction of zeros per feature, and median fraction of zeros per sample. scRNA-seq datasets show extreme sparsity (>89% global zeros), whereas bulk rna-seq datasets show negligible zeros in our processed matrices. Any random subsets can match performance even in scRNA settings where more than 90% zeros exist, meaning the effect is not driven by selecting nonzero columns. This table is now added as Table 4 in Appendix A.
>
> | Dataset | Samples | Features | Global Zero Fraction | Median Feature Zero Fraction | Median Sample Zero Fraction |
> |----------------------------------------|---------|----------|-----------------------|-------------------------------|------------------------------|
> | Qian 2020 (Breast Cancer sc-RNA-Seq)| 15,672  | 22,276   | **89.1%**| 95.8%| 90.6%|
> | Wu 2021 (Breast Cancer sc-RNA-Seq)| 96,088  | 29,733   | **94.1%**| 99.3%| 95.6%|
> | Bischoff 2021 (Lung cancer sc-RNA-Seq)  | 21,052  | 33,514   | **96.4%**| 99.9%| 96.5%|
> | TCGA Pan-cancer (bulk RNA-Seq)| 10,223  | 20,253   | **0.0%**| 0.0%| 0.0%|
>
> **e. Variation in sampled feature fractions**
>
> Ans: We did _not_ tune sampling fractions. For every dataset, our figures include performance curves across the full range of subset sizes. We simply highlight some useful points of comparison in our tables (parameters which have the same feature set size or achieve the same accuracy as the published results)
>
> **f. Interpretation of “outcome edges” in prior papers**
>
> Ans: If we are interpreting this comment correctly: These few outcome edges are chosen from amongst a feature selected subset. If there is a biologically valid outcome edge, that should occur from the first principles. The fact that such an edge appears in the outcome of a computational feature selection algorithm is not significant in practically all the datasets we test - the gap between a cleverly selected feature set and the null hypothesis should be considered when exploring and understanding these outcome edges. Published “outcome edges” are chosen after feature selection; hence if random is able to match FS performance, the biological conclusions need stronger null-model baselines.
>
> We thank the reviewer again for the helpful feedback and hope the clarifications address all concerns.

---

### Author Response · Authors · 2025-11-23
**Summary of changes and responses**

We thank all reviewers for the careful reading and constructive feedback. The reviews collectively raised questions about dataset selection, scope of claims, clarity of experimental procedures, and connections to prior theory.

We have addressed each point in the individual responses and updated the manuscript accordingly.

The revision now includes:
1. a dedicated “dataset inclusion criteria” section 3.2 clarifying selection rules;
2. an explicitly narrowed claim focused on high-dimensional biological datasets;
3. expanded single-cell rna-seq results, including additional large-scale datasets;
4. clearer comparisons between published feature-selected sets, standard fs methods, and random subsets;
5. an updated and clarified description of ensemble experiments;
6. improved tables and figures to make the random-subset baselines and matching-subset sizes explicit;
7. discussion of prior work on signature multiplicity and how our findings relate.

Overall, we hope the updates improve clarity and address the raised concerns.

---

### Meta-Review · Area_Chair_HNGG · 2026-01-04

**Summary:**

Reviewers raised concerns regarding the significance and clarity of the paper. Although most clarity issues have been addressed, the significance of the work remains questionable. Moreover, there are still several outstanding issues in the empirical analysis and presentation. I therefore recommend rejection of the paper.

**Reviewer Concerns:**

Concerns and questions regarding the experimental methodology have been addressed in the rebuttal. However, I believe that the fundamental issue of the paper, its potentially misleading key claim, has not been properly addressed and remains outstanding.

**Reviewer Scores:**

**Reviewer 1ZQK** raised many concerns and questions regarding data processing. However, this reviewer already assigned a score of 10 to the paper. Therefore, the reviewer is likely to maintain this score.

**Reviewer 49XN** raised concerns about details of the experimental protocol. Although the authors provided additional details in their response, the fundamental limitations of the paper remain. Therefore, this reviewer is unlikely to change their score.

**Reviewer AW9D** raised serious concerns regarding the key claims of the paper, which were not properly addressed in the rebuttal. As a result, this reviewer is likely to maintain their original score.

---

### Decision · Program_Chairs · 2026-01-26

Reject